# Equilibrium, Kinetic, and Thermodynamic Studies on Adsorption of Rhodamine B from Aqueous Solutions Using Oxidized Mesoporous Carbons

**DOI:** 10.3390/ma15165573

**Published:** 2022-08-13

**Authors:** Michal Marciniak, Joanna Goscianska, Małgorzata Norman, Teofil Jesionowski, Aleksandra Bazan-Wozniak, Robert Pietrzak

**Affiliations:** 1Faculty of Chemistry, Adam Mickiewicz University in Poznań, Uniwersytetu Poznańskiego 8, 61-614 Poznan, Poland; 2Faculty of Chemical Technology, Poznan University of Technology, Berdychowo 4, 60-965 Poznan, Poland

**Keywords:** ordered mesoporous carbons, hexagonal structure, hard template method, oxidation by nitric acid, cationic dye, adsorption

## Abstract

Oxidized mesoporous carbon C_SBA-15_, obtained by the hard method, was applied to remove rhodamine B from the aqueous system. The process of carbon oxidation was performed using 0.5 and 5 M of nitric (V) acid solution at 70 and 100 °C. Functionalization of mesoporous carbon with HNO_3_ solutions led to reduction in the surface area, pore volume, and micropore area, however, it also led to an increased number of oxygen functional groups of acidic character. The functional groups probably are located at the entrance of micropores, in this way, reducing the values of textural parameters. Isotherms of rhodamine B adsorption indicate that the oxidation of mesoporous carbons resulted in an increase in the effectiveness of the removal of this dye from aqueous solutions. The influence of temperature, pH, and contact time of mesoporous material/rhodamine B on the effectiveness of dye removal was tested. The process of dye adsorption on the surfaces of the materials studied was established to be most effective at pH 12 and at 60 °C. Kinetic studies of the process of adsorption proved that the equilibrium state between the dye molecules and mesoporous carbon materials is reached after about 1 h. The adsorption kinetics were well fitted using a pseudo-second-order model. The most effective in rhodamine B removal was the sample C_SBA-15_-5-100, containing the greatest number of oxygen functional groups of acidic character. The Langmuir model best represented equilibrium data.

## 1. Introduction

Water, which is the element necessary to support all forms of life, covers most of Earth’s surface. Its pollution is negative to human health and the entire hydrosphere [1,2,3,4,5]. There is a constant increase in the content of organic compounds in wastewater, especially in post-production waste produced by different types of industry [6,7], such as the textile industry, reaching several thousand tons a year, of which 10–20% is lost in the process of production [8,9,10]. Dyes are usually well soluble in water and resistant to degradation by various methods (biological, physical, and chemical) [9,10,11,12]. Therefore, they must be removed from the wastewater. At present, a few methods are used to remove organic compounds [12,13,14]. From among newly proposed methods, very attractive seems to be the method of membrane separation, offering simplicity of operation and low energy consumption [15]. On the other hand, membrane efficiency decreases with time because of the fouling of pollutant particles on the membrane surface and inside membrane pores. Moreover, molecules of many organic compounds are smaller than the membrane pores and the process of membrane separation is ineffective [16,17,18]. Currently, the most effective are the adsorption methods that can be used for the removal of organic and inorganic compounds from water solutions. The properties and quality of the adsorbent influence the adsorption efficiency. Many materials, for example, silica gels [19], alumina [20], and zeolite molecular sieves [21,22] are rather ineffective in the adsorption process. Activated carbons are used as organic pollutant adsorbents, although the presence of micropores in their structure limits the penetration of larger dye molecules into the pores. Recently, much attention was paid to ordered mesoporous carbon materials, as they show unique properties, such as thermal and mechanical stability, as well as good textural parameters [23]. These materials are easily modified, which allows for improved adsorption properties towards a range of pollutants [23,24,25,26]. Many methods for carbon material modification were proposed, but the most effective is that of wet oxidation, in which oxygen functional groups are introduced on the carbon surface [27]. A number of authors studied the adsorption of organic compounds on carbon materials, for instance, Liu et al. [28] proposed the removal of Acid Red 73 and Reactive Black 5 by CMK-3, CMK-5, and its carbon/silica composite Si-CMK-5, with different pore structures. The greatest sorption capacity towards these dyes was shown by CMK-5, which was related to double pore systems and large specific surface area. Asouhidou et al. [29] obtained mesoporous carbons, a highly ordered CMK-3 sample with hexagonal structure and a disordered mesoporous carbon (DMC), and tested them in the removal of Remazol Red 3BS, comparing their performance with that of commercial products (Takeda 5A, Calgon, and Norit SAE-2) and a HMS mesoporous silica with a wormhole pore structure. Their results show that the material structure and pore size have a significant impact on the effectiveness of adsorption. The resulting sorption capacities decreased in the order CMK-3 (0.531 mmol/g) > DMC (0.453 mmol/g) > SAE-2 (0.167 mmol/g) > Calgon (0.05 mmol/g) > Takeda 5A = HMS (0.007 mmol/g). The group of Peng [30] successfully applied the ordered mesoporous CMK-3 containing nitrogen functional groups as the adsorbent of Acid Black 1. According to their results, the modification of mesoporous carbon significantly improved its effectiveness in the removal of this dye. The maximum sorption capacities of the initial and functionalized materials were over 270 mg/g and nearly 500 mg/g, respectively.

One of the most popular textile dyes is rhodamine B, which is a cationic synthetic dye of green or red purple crystals [31]. With the release of rhodamine B into the water, several environmental and public health problems are caused [32,33]. Very popular and effective adsorbents of rhodamine B from water solutions are mesoporous carbons [34,35,36]. Therefore the aim of the study was to analyze the process of rhodamine B removal from water solutions by adsorption on oxidized mesoporous carbons of hexagonal structure. The effects of pH of the dye solution, adsorbent/adsorbate contact time, and temperature of the process were checked on the sorption capacity of obtained mesoporous carbons. The mesoporous carbon materials used as adsorbents were characterized in detail by a number of physicochemical methods.

## 2. Materials and Methods

### 2.1. Sample Preparation

#### 2.1.1. Mesoporous Carbon Synthesis

Mesoporous carbon C_SBA-15_ was obtained by the hard template method using the ordered silica SBA-15 as the solid template and sucrose as the carbon precursor [37,38]. The substrates for SBA-15 preparation were 50 mg of Pluronic P123 (EO_20_PO_70_EO_20_; Aldrich, Saint Louis, MO, USA), 19 mL of 1.6 M HCl (Avantor Performance Materials Poland S.A., Gliwice, Poland), and 1.1 mL of TEOS (MERCK KGaA, Darmstadt, Germany). Pluronic P123 was dissolved in a water solution of HCl at 35 °C. To the mixture, TEOS was added dropwise upon stirring continued for 6 h. Then the mixture was subjected to hydrothermal treatment in poly-propylene bottles in a drier (1 day at 35 °C and the next 6 h at 100 °C). Then the material obtained was filtered off and dried (100 °C, 12 h). The template was removed by calcination (550 °C, 8 h). The procedure produced ordered mesoporous silica SBA-15 of hexagonal structure [37,38]. The mesoporous silica SBA-15 was subjected to twice repeated impregnation using sucrose solution. Next, 125 mg sucrose (MERCK KGaA, Darmstadt, Germany) was dissolved in H_2_SO_4_ (0.14 mL, Avantor Performance Materials Poland S.A., Gliwice, Poland) and 5 mL of distilled water. The solution was added to the flask with the ordered silica. The contents were heated in the oven (6 h at 100 °C and 6 h at 160 °C). The obtained material was subjected to another impregnation with a solution of sucrose (800 mg), H_2_SO_4_ (0.09 mL), and distilled water (5 mL).

The composite was then subjected to carbonization by heating for 2 h at 900 °C. The remaining silica was washed out twice with 5% of HF solution (200 mL, Avantor Performance Materials Poland S.A., Gliwice, Poland). The material was filtered off, washed with C_2_H_5_OH, and dried (12 h at 100 °C) [39].

#### 2.1.2. Sample Functionalization

The C_SBA-15_ of the hexagonal structure was subjected to oxidation at 70 or 100 °C with the use of HNO_3_ at the concentrations of 0.5 or 5 mol/L as the oxidizing agent. Next, 0.5 g of the carbon was placed in a round-bottomed flask and flooded with 30 mL of nitric (V) acid solution. Oxidation was performed under reflux (12 h). Next, the contents were filtered off and the carbon was washed with C_2_H_5_OH and distilled water. The oxidized carbon materials were labeled as C_SBA-15_-0.5-70, C_SBA-15_-5-70, C_SBA-15_-0.5-100, and C_SBA-15_-5-100, where 70 and 100 refer to the temperature of oxidation, while 0.5 and 5 to the concentration of HNO_3_.

### 2.2. Analytical Procedures

The texture parameters of the samples obtained were characterized by low-temperature nitrogen adsorption/desorption isotherms measured on a sorptometer Quantachrome AutosorbiQ (Boynton Beach, FL, USA) [9].

X-ray diffraction patterns were obtained on a Bruker AXS DB Advance diffractometer (CuK_α_ radiation, λ = 0.154 nm, step size 0.02°).

The number of surface oxygen functional groups was determined by the Boehm method [40]. A portion of 0.25 g of adsorbent was placed in 25 mL of 0.1 mol/L solutions of either NaOH or HCl. The vials were sealed and shaken for 24 h and then 10 mL of each filtrate was pipetted and the excess of base or acid was titrated with 0.1 mol/L HCl or NaOH, as required. The numbers of acidic sites of various types were calculated assuming that NaOH neutralizes all acidic groups and HCl reacts with all basic groups.

Structural changes in the oxidized mesoporous carbon materials were determined by FT-IR spectroscopy. The preparation of samples is described in the paper [9]. The study was carried out on a Varian 640-IR spectrometer (Agilent, Santa Clara, CA, USA).

Zeta potential was determined using a Zetasizer Nano ZS instrument equipped with an autotitrator (Malvern Instruments Ltd., Malvern, United Kingdom) [41]. The electrophoretic mobility of the particles was measured and converted to the zeta potential according to the Henry Equation (1):(1)$$R_{ef}=\frac{2\varepsilon \zeta f\left(K_{a}\right)}{3\eta }$$
where *R_ef_* is the electrophoretic mobility, ε the dielectric constant; *ζ* the electrokinetic (zeta) potential; *η* the viscosity, and f (*K*_a_) the Henry function. The isoelectric point is a pH value at which zeta potential is zero, the surface has net electrical neutrality. When pH > i_ep_, the surface charge is negative and pH < i_ep_, it is positive.

### 2.3. Adsorption of Rhodamine B

The carbon material portions of 0.02 g were placed in flasks and flooded with 50 mL of a dye solution of a given concentration (25–250 mg/L), and the contents were shaken at 22 ± 1 °C for 24 h. Spectrophotometric measurements were carried out with a spectrometer Cary 100 Bio (Agilent, Santa Clara, CA, USA). Rhodamine B absorbs the irradiation of λ_max_ = 553 nm. The amount of rhodamine B adsorbed on the oxidized mesoporous carbons was calculated from Equation (2):(2)$$q_{e}=\frac{C_{0}-C_{e}}{m}\times V$$
where: *C*_0_—initial rhodamine B concentration (mg/L); *C_e_*—equilibrium rhodamine B concentration (mg/L); *m*—the mass of mesoporous carbon sample (g); *V*—volume of rhodamine B solution (L). The experimental adsorption studies were carried out twice and are shown with a standard deviation error.

The effects of pH (CP-401 pH-meter, ELMETRON, Zabrze, Poland) of the dye solutions, temperature, and contact time of the sample/rhodamine B on the sorption capacities of mesoporous carbons were studied.

### 2.4. Adsorption Modeling

Experimental data were fitted to pseudo-first-order (3) and pseudo-second-order (4) models [9]:(3)$$ln\left(q_{e}-q_{t}\right)=lnq_{e}-\frac{k_{1}t}{2.303}$$
where: *q_e_*—sorption capacity of the rhodamine B adsorbed at equilibrium state (mg/g); *q_t_* —sorption capacity of the rhodamine B adsorbed in time (mg/g); and *k*_1_—the rate constant for the pseudo-first order model (min^−1^).

The pseudo-second-order model can be expressed by the equation:(4)$$\frac{t}{q_{t}}=\frac{1}{k_{2}q_{e}{}^{2}}+\frac{t}{q_{e}}$$
where: *k*_2_—the rate constant for pseudo-second order model (g/mg·min).

### 2.5. Thermodynamic Study

Thermodynamic parameters [42,43,44] were calculated by using the following Equation (5):(5)$$\Delta G^{0}=-RTlnK_{d}$$
where: Δ*G*^0^—Gibbs free energy (J·mol^−1^); *R*—universal constant (8.314 J·mol^−1^·K^−1^); *T* temperature (K); and *K_d_*—thermodynamic equilibrium constant.

The Gibbs free energy of adsorption (Δ*G*^0^) can be represented by the Equation (6):(6)$$\Delta G^{0}=\Delta H^{0}-T\Delta S^{0}$$
where: Δ*H*^0^—enthalpy change; Δ*S*^0^—entropy change.

Thermodynamic parameters can be also calculated from Equation (7):(7)$$lnK_{d}=\frac{\Delta S^{0}}{R}+\frac{\Delta H^{0}}{RT}$$

Δ*H*^0^ and Δ*S*^0^ parameters were calculated (7) from the slope and intercept of the plot of *lnK_d_* versus 1/T yields, respectively.

### 2.6. Adsorption Isotherms

In our work, we used the Langmuir and Freundlich models to explain the mechanism of rhodamine B adsorption on oxidized carbon materials [45,46].

The linear equation of Langmuir isotherm (8) is represented as follows:(8)$$\frac{C_{e}}{q_{e}}=\frac{1}{K_{L}\times q_{max}}+\frac{C_{e}}{q_{max}}$$
where: *C_e_*—equilibrium concentration of rhodamine B (mg/L); *q_e_*—sorption capacity of rhodamine B adsorbed onto the adsorbent at equilibrium (mg/g); *q_m_*—maximum monolayer adsorption capacity of adsorbent (mg/g); and *K_L_*—Langmuir adsorption constant (L/mg).

The Freundlich isotherm is expressed mathematically as [46] (9):(9)$$lnq_{e}=lnK_{F}+\frac{1}{n}lnC_{e}$$
where: *q_e_*—sorption capacity of rhodamine B adsorbed at equilibrium (mg/g); *C_e_*—equilibrium concentration of rhodamine B (mg/L); *K_F_*—Freundlich adsorption constant (mg/g·(L/mg)^1/n^); and *n*—Freundlich constant indicates how favorable the adsorption process is.

Experimental data were also fitted to the non-linear Langmuir and Freundlich models.

## 3. Results and Discussion

### 3.1. Characterization of Adsorbents

The data on textural parameters are presented in Table 1. The sample of C_SBA-15_ is characterized by a well-developed surface area (S_BET_ = 1203 m^2^/g) and a total pore volume (V_t_) of 1.32 cm^3^/g. The carbon obtained in this way also has micropores, whose area (S_micro_) is 682 m^2^/g. The process of functionalization of sample C_SBA-15_ leads to a decrease in these parameters. However, the changes are not uniform and depend on the conditions of the oxidation process. Although sample C_SBA-15_-5-100 was obtained by treatment with a 5 mol/L solution of HNO_3_ at 100 °C, it shows the largest surface area from among all oxidized materials (854 m^2^/g). The micropore area and total pore volume of this material are 198 m^2^/g and 1.10 cm^3^/g, respectively. Most probably in the conditions applied (100 °C, 5 mol/L HNO_3_), the micropores are unblocked again and the carbon compounds are removed, which increases the micropore area and total surface area [47]. The smallest S_BET_ of 685 m^2^/g, S_micro_ of 106 m^2^/g, and V_t_ of 0.89 cm^3^/g were obtained for sample C_SBA-15_-5-70. The reduced values of textural parameters are most probably caused by the localization of the newly generated oxygen functional groups in micropores, which leads to their blocking and decreases the surface area.

Recorded in the small angle range XRD, diffractograms of the adsorbents obtained are depicted in Figure 1. Diffractograms of all samples show one intensive peak characteristic of hexagonal pore arrangement [9]. The diffractograms of the initial material C_SBA-15_ and samples C_SBA-15_-0.5-100 and C_SBA-15_-5-100 contain also the reflections in the range 2Θ ≈ 1.7–2.5°, corresponding to the planes (100), (110), and (200) of P6 mm structure, evidencing good ordering of the materials. No analogous reflections are observed for C_SBA-15_-0.5-70 and C_SBA-15_-5-70, which indicates a partial disturbance in the mesoporous structure.

Table 2 presents the results obtained from Boehm titration. The oxidation of mesoporous sample C_SBA-15_ with a solution of HNO_3_ generates on its surface acidic functional groups whose content depends on the conditions of the process. When using 0.5 mol/L and 5 mol/L HNO_3_ solution, the content of basic groups decreased to 0.13 mmol/g for samples C_SBA-15_-0.5-70 and C_SBA-15_-0.5-100, and to their total disappearance on samples C_SBA-15_-5-70 and C_SBA-15_-5-100. For the acidic functional groups, the results are different. The content of acidic groups in the initial carbon material C_SBA-15_ is 1.09 mmol/g. The content of such groups increases after oxidation and clearly depends on the functionalization conditions, which are the concentration of nitric (V) acid and process temperature. The highest content of the acidic functional groups of 4.88 mmol/g was observed for the sample oxidized with 5 mol/L HNO_3_ solution at 100 °C (C_SBA-15_-5-100). Reduction in the temperature of the process or in the concentration of the oxidizing agent led to lower content of acidic surface functional groups: C_SBA-15_-0.5-70–2.14 mmol/g, C_SBA-15_-5-70–3.24 mmol/g, and C_SBA-15_-0.5-100–3.45 mmol/g.

Figure 2 presents the transmission FT-IR spectra of the initial carbon C_SBA-15_ and oxidized samples of acidic surface nature.

The FT-IR spectra show a rather broad band at about 1200 cm^−1^ that can be assigned to the stretching vibrations of the C–O bond in ethers, acid anhydrides, or phenol. The bands about 1600 cm^−1^ correspond to the stretching vibrations of carbon–carbon bonds in the aromatic ring. There is also a clearly visible band at 3400 cm^−1^, assigned to the stretching vibrations of O–H bonds in hydroxyl groups, whose presence can indicate the oxidation of mesoporous carbon surface and the presence of carboxyl or phenol oxygen groups [48,49]. For the oxidized materials, a new band appeared at about 1750 cm^−1^, which was most pronounced for C_SBA-15_-5-100. This band can be assigned to the carbonyl group of aldehyde, ester, or carboxyl acid. The most probable origin of this band is the presence of -COOH groups on the surface of oxidized carbons, which is supported by the simultaneous presence of a band at about 3400 cm^−1^ [48,49]. The bands of the highest intensity were obtained for sample C_SBA-15_-5-100, which has the highest content of functional acidic groups. The bands recorded for the other samples were less intensive because of the lower content of functional groups.

### 3.2. Adsorption Studies

In this work, we studied the kinetics of adsorption (Figure 3). The adsorption of rhodamine B on the surface of the carbon materials studied is very fast for the first 10 min. After this time, the majority of the active sites on the carbon surface is already occupied by the dye molecules and the rate of adsorption considerably decreases. After 1 h, no increase *q_e_* was noted, which means that a state of equilibrium was reached and there are no more active sites on the carbon material studied. Next step, the experimental data were fitted to two kinetic models: pseudo-first-order (PFO) and pseudo-second-order models (PSO).

According to data collected in Table 3, the R^2^ for the PFO model takes values from the range 0.9265 to 0.9865. Much higher, and the same for all carbon material values of R^2^ (0.9999), were obtained assuming the PSO model. Moreover, the sorption capacities calculated assuming this model are mostly in agreement with their experimental values. Therefore, we conclude that the kinetics of rhodamine B adsorption on the surface of the samples studied can be described by the PSO model.

Figure 4 illustrates the effect of the adsorbate solution pH, changed in the range 2–12, on the amount of adsorbed rhodamine B, while Figure 5A presents the zeta potential curves vs. pH of mesoporous carbons before the dye adsorption, and Figure 5B, after its adsorption. As shown in Figure 5A, sample C_SBA-15_ has the isoelectric point (i_ep_) at 3.4. The functionalization changes the adsorbents’ surface, for C_SBA-15_-5-70 the isoelectric point is at pH 2.6 and for C_SBA-15_-5-100 it does not exist; the zeta potential in the whole range of measurement is negative. The negative zeta potential indicates that oxidation treatment with HNO_3_ introduces hydroxyl, carboxylic, and carbonyl groups on the surfaces of the samples, which dissociate generating the negatively charged surface [50,51]. The zeta potential value higher than 25 mV, positive or negative, is indicative of electrokinetic stability [52]. Pristine carbon sample C_SBA-15,_ and the samples after oxidation (C_SBA-15_-5-70 and C_SBA-15_-5-100), show good electrokinetic stability at pH values higher than 6 (Figure 5A). The data presented in Figure 4 suggest that the effectiveness of rhodamine B removal from water solutions depends on the pH of the adsorbate solution. The lowest sorption capacity was recorded at pH 2, which can be explained as a result of the protonation of carboxyl, amide, and hydroxyl groups, leading to the generation of a positive charge, which is engaged in repulsive interaction with the positive charge of the cationic dye (rhodamine B). In addition, at low pH, the H^+^ cations compete with rhodamine B cations for the adsorption sites. The surface is more negatively charged when the pH increases as a result of deprotonation of the functional groups by the hydroxide anion OH-, which is more favorable for the adsorption of rhodamine B. Then, we have strong electrostatic attraction between the negatively charged surface of the adsorbent and the positively charged cationic dye. At pH 12, the sorption capacities of all adsorbents increased by about 20 mg/g in relation to that of the initial carbon sample. The zeta potential values of C_SBA-15_-5-100 after adsorption of rhodamine B from the solutions of the concentration of 50 mg/L and 150 mg/L (Figure 5B) differ from those for C_SBA-15_-5-100 before adsorption, and i_ep_ can be observed (changes in i_ep_ value suggest the chemical adsorption) [53].

The effect of temperature on the amount of rhodamine B adsorbed on the surface of the samples studied was also checked (Figure 6). The amount of rhodamine B adsorbed (*q_e_*) on the surface of mesoporous carbon materials before and after their oxidation increases with temperature. It follows the consequence of increased mobility of dye molecules at higher temperatures. For instance, the amount of organic dye adsorbed on the surface of sample C_SBA-15-_5-100 is 227 mg/g (25 °C), 254 mg/g (45 °C), and 261 mg/g (60 °C).

According to the data collected in Table 4, the results of ΔG° prove that the process of adsorption of dye is spontaneous. For all samples, the degree of spontaneity was the highest at 60 °C. The positive values of ΔH obtained reveal an endothermic adsorption process. In addition, the positive values of ΔSevidence increased the degree of randomness at the interface in the process of the rhodamine B adsorption.

Figure 7 presents the isotherms of dye adsorption on the surface of carbon adsorbents. According to the results, the initial carbon C_SBA-15_ proved to be the least effective in the removal of dye, despite having the largest surface area from among all samples studied (1203 m^2^/g). However, this material also has the lowest number of surface functional groups of acidic character. Further data analysis allows us to conclude that the process of the adsorption of organic dye depends on surface functional groups of acidic character. Sample C_SBA-15_-5-100, showing the largest sorption capacity towards rhodamine B (325 mg/g), has the greatest content of such functional groups (4.88 mmol/g). The process of its oxidation was performed with 5 mol/L nitric (V) acid solution at 100 °C. The amounts of rhodamine B adsorbed on the other oxidized carbon materials C_SBA-15_-0.5-70, C_SBA-15_-5-70, and C_SBA-15_-0.5-100 were 248, 268, and 302 mg/g, respectively.

Analysis of the adsorption data was performed with the use of two theoretical models of Langmuir and Freundlich (Figure 8 and Figure 9, Table 5 and Table 6). Comparison of the experimental data with the predictions of a particular model provides information on the mechanism of adsorption and mesoporous carbon/rhodamine B interactions. The criterion of best fitting was the determination coefficient R^2^.

The determination coefficient R^2^ for the linear form of the Langmuir model takes values from 0.9987 to 0.9999 for all adsorbents obtained. The values of R^2^ (Freundlich model) for particular carbon materials were: 0.8564 for C_SBA-15_, 0.7993 for C_SBA-15_-0.5-70, 0.8009 for C_SBA-15_-5-70, 0.8611 for C_SBA-15_-0.5-100, and 0.8598 for C_SBA-15_-5-100. Therefore, the isotherms of rhodamine B adsorption on the surface of the carbon materials studied correspond to the Langmuir model. It should be noted that the experimental values of *q_e_* are slightly lower from the theoretical *q_m_*. For the oxidized mesoporous carbons, the coefficient *K_L_* values are higher than for the initial material C_SBA-15_, which means that the bonding between rhodamine B and the functionalized carbon material surfaces is stronger. The coefficient 1/n for the Freundlich model lies between (0 < 1/n < 1) which shows that this isotherm is favorable.

Additionally, the comparison of the non-linear form of the Langmuir and the Freundlich [54] isotherm models with experimental data for the adsorption of rhodamine B on the surface of carbon adsorbents is presented in Table 6 and Figure 9. It was established that the Langmuir adsorption model indicates a better fit to the experimental data than the Freundlich model. The R^2^ values for the non-linear and linear Langmuir isotherms were similar.

Table 7 presents a comparison of the sorption capacities of carbon samples obtained towards tested dye with those of other adsorbents. The comparison implies that the ordered mesoporous carbon materials are very effective in the removal of rhodamine B from water solutions. The majority of literature reported adsorbents, including hierarchical SnS_2_ nanostructure [5], TA-G [55], iron-pillared bentonite [56], sago waste activated carbon [57], kaolinite [58], [Ni_(bipy)2_]_2_(HPW_12_O_40_) [59], orange peel [60], whose sorption capacities are lower than for the oxidized carbon materials studied in this work. From among all materials mentioned in Table 6, the highest sorption capacity towards rhodamine B showed the magnetic mesoporous carbon materials (342–400 mg/g) [61], also high sorption capacities were noted for MoS_2_/MIL-101-345 mg/g [62] and oxidized ordered mesoporous carbon material C_SBA-15_-5-100 (325 mg/g). The hierarchical SnS_2_ nanostructure [5] and TA-G [55] adsorption capacity was at a level of 200 mg/g, and the sorption capacities of the other samples listed in Table 7 were much lower: iron-pillared bentonite—99 mg/g [56], sago waste activated carbon—47 mg/g [57], kaolinite—46 mg/g [58], [Ni_(bipy)2_]_2_(HPW_12_O_40_)—23 mg/g [59], and orange peel—14 mg/g [60].

## 4. Conclusions

According to the above-presented results, functionalization of the ordered mesoporous carbon of hexagonal structure (C_SBA-15_) using a solution of nitric (V) acid, has brought about an increase in its effectiveness of rhodamine B removal from water solutions. Oxidization of mesoporous carbon reduced its textural parameters but increased the acidic character of the surface. Moreover, for the samples functionalized at 70 °C, the oxidation resulted in the disappearance of the reflections corresponding to the carbon structure ordering, as confirmed by XRD diffractograms in the small angle range. Sample C_SBA-15_-5-100 was characterized by the highest content of surface functional groups of acidic nature and the largest surface area from among the functionalized carbon materials studied. This sample was the most effective in the removal of tested dye from water solutions (325 mg/g). The sorption capacity towards rhodamine B depended also on the pH of the solution and the process temperature. The adsorption of organic dye was more effective at higher pH because of deprotonation of the surface functional groups on the carbon samples by the hydroxide anion. The measurements of zeta potential, before and after adsorption, proved that with increasing pH, the surface charge on carbon samples changed. The amount of adsorbed rhodamine B was also found to increase with the increasing temperature of the process, which is related to increased mobility of the dye molecules at higher temperatures. The thermodynamic parameters showed that the adsorption process of dye was endothermic and proceeded spontaneously. The adsorption of rhodamine B onto oxidized mesoporous carbons was described by the Langmuir isotherm and pseudo-second-order kinetic model.

## Figures and Tables

**Figure 1 materials-15-05573-f001:**
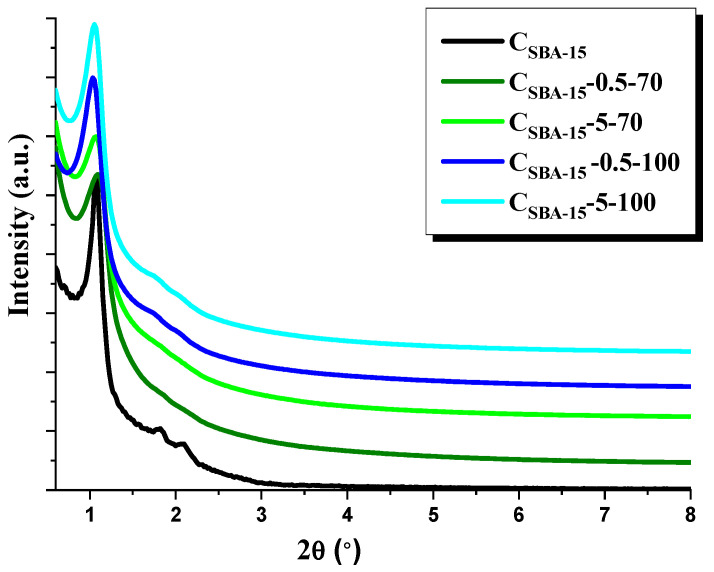
XRD patterns of adsorbents obtained.

**Figure 2 materials-15-05573-f002:**
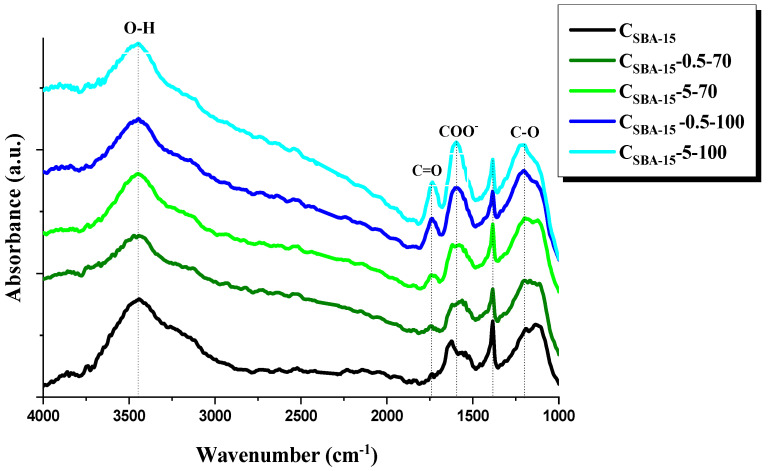
FT-IR spectra of carbon adsorbents obtained.

**Figure 3 materials-15-05573-f003:**
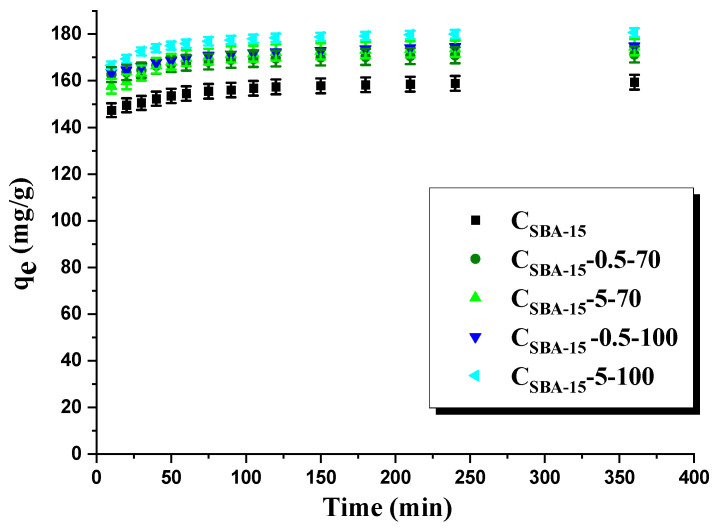
Amount of rhodamine B adsorbed on adsorbents obtained (*C*_0_–75 mg/L).

**Figure 4 materials-15-05573-f004:**
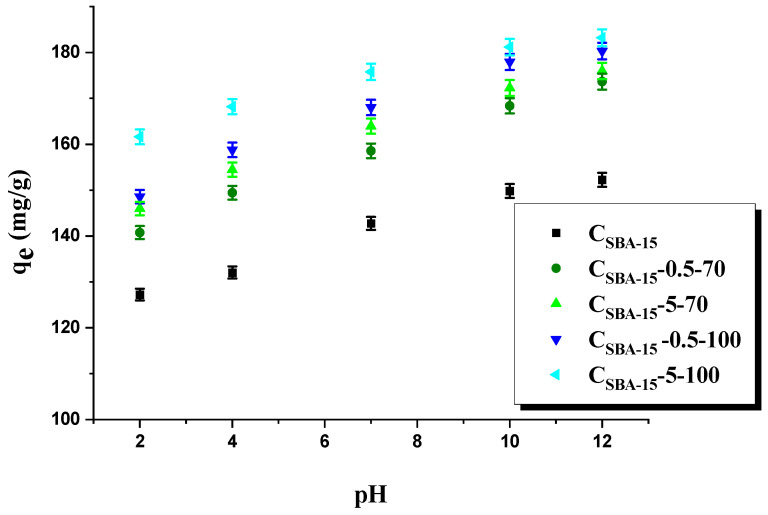
The pH influence on the sorption capacity of adsorbents obtained.

**Figure 5 materials-15-05573-f005:**
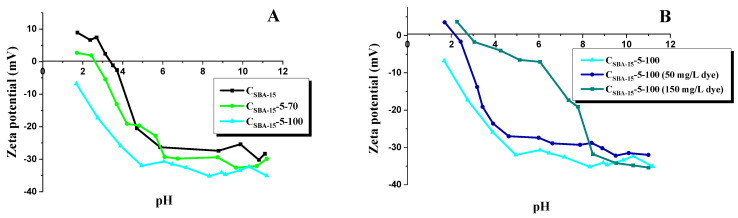
Zeta potential curves vs. pH for C_SBA-15_, C_SBA-15_-5-70, and C_SBA-15_-5-100 mesoporous carbons before dye adsorption (**A**) and for C_SBA-15_ and C_SBA-15_-5-100 after adsorption of rhodamine B (**B**).

**Figure 6 materials-15-05573-f006:**
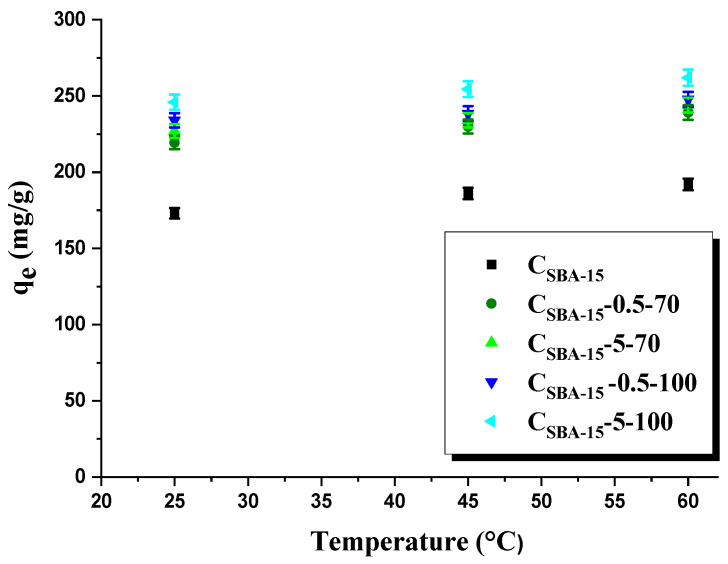
Effect of temperature on the adsorption of dye (100 mg/L) onto adsorbents obtained.

**Figure 7 materials-15-05573-f007:**
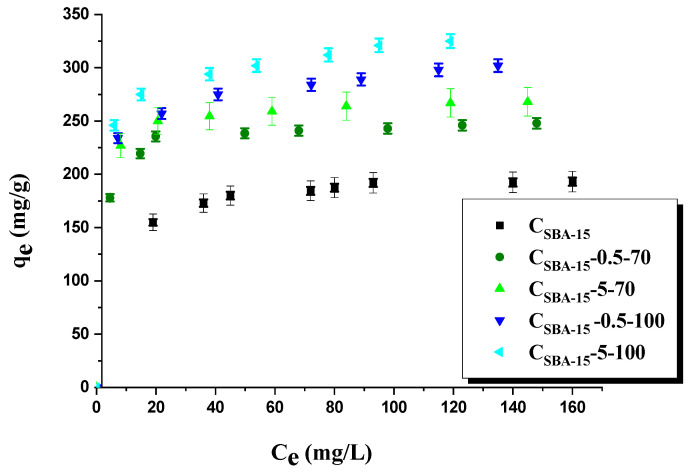
Adsorption of rhodamine B onto adsorbents obtained.

**Figure 8 materials-15-05573-f008:**
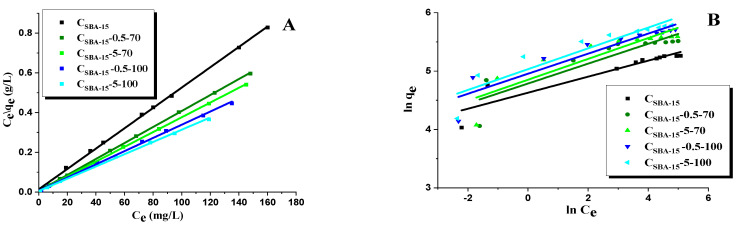
Linear fitting of rhodamine B adsorption isotherms onto carbon adsorbents to Langmuir (**A**) and Freundlich (**B**) models.

**Figure 9 materials-15-05573-f009:**
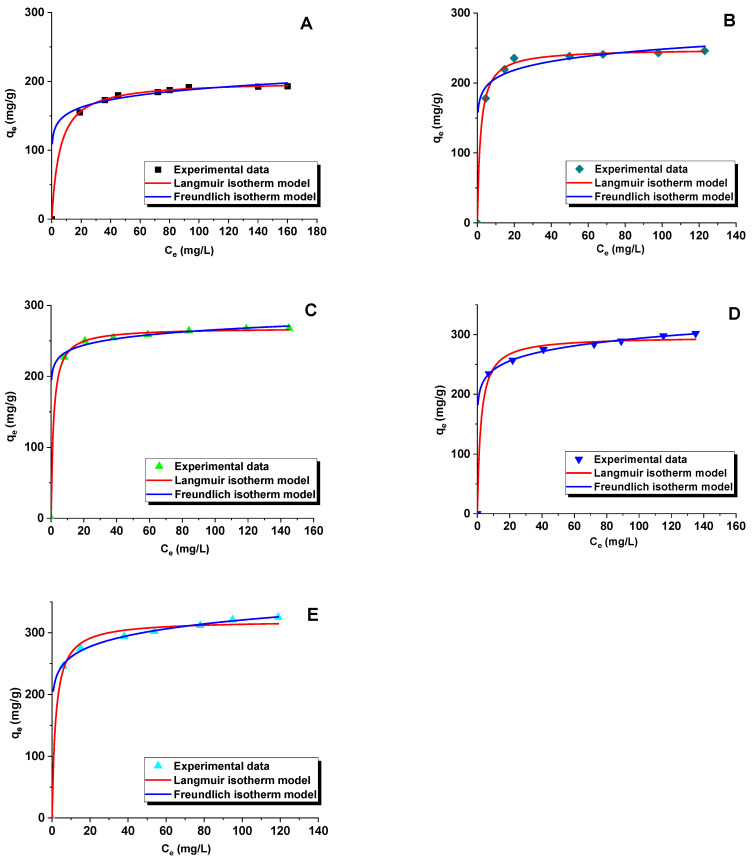
Non-linear fitting of rhodamine B adsorption isotherms to Langmuir and Freundlich models for samples: C_SBA-15_ (**A**), C_SBA15_-0.5-70 (**B**), C_SBA-15_-5-70 (**C**), C_SBA-15_-0.5-100 (**D**), and C_SBA-15_-5-100 (**E**).

**Table 1 materials-15-05573-t001:** Textural parameters of adsorbents obtained ^1^.

Adsorbent	S_BET_ (m^2^/g)	V_t_ (cm^3^/g)	S_micro_ (m^2^/g)	Average Pore Diameter (nm)
C_SBA-15_	1203	1.32	685	4.45
C_SBA-15_-0.5-70	789	1.12	147	5.66
C_SBA-15_-5-70	685	0.89	106	5.18
C_SBA-15_-0.5-100	837	1.04	204	4.91
C_SBA-15_-5-100	854	1.10	198	5.18

^1^ Error range between 2% and 5%.

**Table 2 materials-15-05573-t002:** The results obtained from Boehm titration.

Adsorbent	Acidic Groups (mmol/g)	Basic Groups (mmol/g)	Total Content of Acidic and Basic Groups (mmol/g)
C_SBA-15_	1.09 ± 0.01	0.74 ± 0.01	1.83
C_SBA-15_-0.5-70	2.14 ± 0.02	0.13 ± 0.01	2.27
C_SBA-15_-5-70	3.24 ± 0.02	0.00 ± 0.00	3.24
C_SBA-15_-0.5-100	3.45 ± 0.02	0.13 ± 0.01	3.58
C_SBA-15_-5-100	4.88 ± 0.03	0.00 ± 0.00	4.88

**Table 3 materials-15-05573-t003:** Kinetic models parameters.

Adsorbent	*q_e_* (mg/g)	PFO Model			PSO Model		
		*q_e_*_[cal]_ (mg/g)	*k*_1_ (min^−1^)	R^2^	*q_e_*_[cal]_ (mg/g)	*k*_2_ (g/mg min)	R^2^
C_SBA-15_	159.41 ± 3.19	9.94	0.019	0.9391	161.29	0.003	0.9999
C_SBA15_-0.5-70	171.29 ± 3.43	7.35	0.018	0.9265	172.41	0.004	0.9999
C_SBA-15_-5-70	174.46 ± 3.49	11.93	0.022	0.9253	175.44	0.003	0.9999
C_SBA-15_-0.5-100	174.90 ± 3.49	11.44	0.026	0.9865	175.45	0.003	0.9999
C_SBA-15_-5-100	180.71 ± 3.61	10.72	0.023	0.9687	181.81	0.003	0.9999

**Table 4 materials-15-05573-t004:** Thermodynamic parameters obtained.

Adsorbent	Temperature (°C)	∆*G*^0^ (kJ·mol^−1^)	∆*H*^0^ (kJ·mol^−1^)	∆*S*^0^ (J·mol^−1^·K^−1^)
C_SBA-15_	25	−16.41	31.20	159.71
	45	−19.53		
	60	−22.01		
C_SBA-15_-0.5-70	25	−19.51	23.96	145.66
	45	−22.18		
	60	−24.51		
C_SBA-15_-5-70	25	−22.27	22.44	149.90
	45	−25.14		
	60	−27.53		
C_SBA-15_-0.5-100	25	−19.12	43.69	210.76
	45	−23.24		
	60	−26.51		
C_SBA-15_-5-100	25	−21.66	47.17	230.88
	45	−26.19		
	60	−29.73		

**Table 5 materials-15-05573-t005:** The parameters of linear form of Langmuir and Freundlich models for rhodamine B adsorption onto carbon materials.

Material	Langmuir			Freundlich		
	*q_m_* (mg/g)	*K_L_* (L/mg)	R^2^	*K_F_* (mg/g (L/mg)^1/n^)	1/n	R^2^
C_SBA-15_	196.07	0.369	0.9999	102.48	0.1373	0.8564
C_SBA15_-0.5-70	250.00	0.727	0.9998	119.92	0.1701	0.7993
C_SBA-15_-5-70	270.27	0.787	0.9998	127.98	0.1783	0.8009
C_SBA-15_-0.5-100	303.03	0.493	0.9988	141.83	0.1717	0.8611
C_SBA-15_-5-100	322.58	0.574	0.9987	153.34	0.1780	0.8598

**Table 6 materials-15-05573-t006:** The parameters of non-linear Langmuir and Freundlich models for rhodamine B adsorption onto carbon materials.

Material	Langmuir			Freundlich		
	*q_m_* (mg/g)	*K_L_* (L/mg)	R^2^	*K_F_* (mg/g (L/mg)^1/n^)	1/n	R^2^
C_SBA-15_	200.64	0.179	0.9999	122.13	0.0949	0.8983
C_SBA15_-0.5-70	248.73	0.566	0.9986	171.94	0.0793	0.8053
C_SBA-15_-5-70	268.28	0.660	0.9995	208.07	0.0530	0.9357
C_SBA-15_-0.5-100	296.66	0.454	0.9942	197.63	0.0860	0.9953
C_SBA-15_-5-100	319.83	0.505	0.9950	212.12	0.0897	0.9922

**Table 7 materials-15-05573-t007:** Comparison of sorption capacities of oxidized mesoporous carbons with other adsorbents presented in literature.

Adsorbent	Sorption Capacity (mg/g)	References
oxidized mesoporous carbon	248–325	This study
hierarchical SnS_2_ nanostructure	200	[5]
TA-G	201	[55]
iron-pillared bentonite	99	[56]
sago waste activated carbon	47	[57]
kaolinite	46	[58]
[Ni_(bipy)2_]_2_(HPW_12_O_40_)	23	[59]
orange peel	14	[60]
magnetic mesoporous carbon materials	342–400	[61]
MoS_2_/MIL-101	345	[62]

## Data Availability

Data is contained within the article.

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
