# Peer review of "Equilibrium, Kinetic, and Thermodynamic Studies on Adsorption of Rhodamine B from Aqueous Solutions Using Oxidized Mesoporous Carbons"

_materials, 2022, doi:10.3390/ma15165573_

Round 1
Reviewer 1 Report
I do confirm the analytical results done by the authors. This manuscript is well structured within the quality of materials. The adsorption kinetic and mineralogical results are valuable to other researchers. However, some minor modifications should be finished. For example, the literature review regarding oxidized mesoporous carbons and the toxic effect of Rhodamine B in the environment should be improved.
1. Please add the literature review regarding oxidized mesoporous carbons and the toxic effect of Rhodamine B in the environment 2. In Table 2, how did authors compare the increased values of total content of acidic and basic groups? 3. In Figure 3, please explain the reason that high adsorption capacity at the 0 min.Author Response
The answers to the referee as well as a detailed list of changes introduced are included.

Reviewer 2 Report
In this work a novel nanoadsorbent materials based on mesoporous carbon were prepared, and studied for adsorption of organic pollutant Rhodamine B dye from aqueous solutions. The synthesis and characterization of the materials are described with great accuracy, and the sorption capacities of the materials are found to be superior or competitive compared to those of several other materials described in the literature.
The paper is interesting and the research is done sufficiently accurately, therefore I can recommend its publication after some improvements.
Major remarks.
1. The continuous lines in Figure 7 are misleading, as they are probably splines produced by the plotting software.
Please fit the data to the non-linear Langmuir model and plot the data together with the fitted theoretical lines.
Minor remarks.
1. Figures should be replaced by high quality / resolution figures.
2. The third sentence of the Abstract is hard to understand, a revision may be needed.
3. lines 230-231: “ relatively low intensive” is probably not an accurate description.
4. Please explain (in the manuscript) how were the error bars determined on the isotherm and kinetic graphs.
5. Reference 40 for Freundlich’s original paper is not accurate.
Freundlich, Herbert. "Über die Adsorption in Lösungen" Zeitschrift für Physikalische Chemie, vol. 57, no. 1, 1907, pp. 385-470. https://doi.org/10.1515/zpch-1907-5723
Author Response
The answers to the referee as well as a detailed list of changes introduced are included.

Round 2
Reviewer 2 Report
Authors adressed properly the comments of the reviewer. The paper can be recommended for publication, though a few mistakes can still be spotted by careful reading.
Minor issues
1. Lines 176-177: an additional sentence seems needed for keeping a coherent description of the models.
2. In references, the journal name abbreviations are not always correct. All of them need to be checked carefully.
I found mistakes in refs 22,37,39,46,47
Author Response
The answers to the referee (Round 2) as well as a detailed list of changes introduced are included.
